# Two StAR-related lipid transfer proteins play specific roles in endocytosis, exocytosis, and motility in the parasitic protist *Entamoeba histolytica*

**Koushik Das**, **Natsuki Watanabe**, **Tomoyoshi Nozaki**\*

Department of Biomedical Chemistry, Graduate School of Medicine, The University of Tokyo, Tokyo, Japan

\* nozaki@m.u-tokyo.ac.jp

**Data Availability Statement:** All relevant data are within the manuscript and its Supporting Information files.

## Abstract

Lipid transfer proteins (LTPs) are the key contributor of organelle-specific lipid distribution and cellular lipid homeostasis. Here, we report a novel implication of LTPs in phagocytosis, trogocytosis, pinocytosis, biosynthetic secretion, recycling of pinosomes, and motility of the parasitic protist *E. histolytica*, the etiological agent of human amoebiasis. We show that two StAR-related lipid transfer (START) domain-containing LTPs (named as EhLTP1 and 3) are involved in these biological pathways in an LTP-specific manner. Our findings provide novel implications of LTPs, which are relevant to the elucidation of pathophysiology of the diseases caused by parasitic protists.

## Author summary

We showed that EhLTP1, but not EhLTP3, is involved in secretion of cysteine protease, the well-established degrading factor of host cells and the extracellular matrix, and in pseudopod formation and migration. In contrast, EhLTP3, but not EhLTP1, is exclusively involved in pinocytosis of the fluid-phase marker. Both EhLTP1 and EhLTP3 are also involved in trogocytosis (ingestion by nibbling) of live mammalian cells and phagocytosis of dead cells. In trogocytosis and phagocytosis, these two LTPs displayed distinct patterns of recruitment: e.g., EhLTP1 was associated at the ligand attachment site at the initiation of trogocytosis, followed by the recruitment of EhLTP3 onto the "trogocytic tunnel" at the intermediate stage of trogocytosis before the closure of the trogosome. Such tempo-spatially coordinated involvement of LTPs in the course of trogo- and phagocytosis has never been demonstrated in unicellular eukaryotes. Neither has LTP been shown to be involved in both endocytosis and exocytosis.

## Introduction

Eukaryotic cells are organized into separate membrane-bound compartments that have unique biochemical signature and specialized function [1]. Maintenance and regulation of specialized

**Funding:** This work was supported partly by TBRF postdoctoral fellowship from The Tokyo Biochemical Research Foundation (TBRF) to K.D. (TBRF-RF17-105), Grant-in-Aid for Scientific Research (B) (JP18H0265, JP21H02723) to T.N. and Grant-in-Aid for Young Scientists (Start-up) and (B) (JP20K22758, JP21K15426) to N.W. from the Japan Society for the Promotion of Science, and Grant for research on emerging and re-emerging infectious diseases from Japan Agency for Medical Research and Development (AMED, JP19fk0108046 and JP20fk0108138) to T.N.), Grant for Science and Technology Research Partnership for Sustainable Development (SATREPS) from AMED and Japan International Cooperation Agency (JICA) (JP19jm0110009 and JP20jm0110022) to T.N.). The funders had no role in study design, data collection and analysis, decision to publish, or preparation of the manuscript.

**Competing interests:** The authors have declared that no competing interests exist.

identity of each compartment are governed by the uneven distribution and intra-cellular movement of two essential biomolecules, proteins and lipids [2]. In contrast to proteins which are precisely distributed to cell organelles either by their intrinsic targeting motifs or through post-translational modifications, lipids lack any such specific targeting motifs that mediate their distinct intracellular distribution [2]. Non-vesicular lipid transport facilitated by lipid transfer proteins (LTPs), are the key contributor of organelle-specific lipid distribution and cellular lipid homeostasis [3]. While the functional implication of lipid transport in organelle-specific lipid distribution, cellular lipid homeostasis and related physiological disorder have been studied mostly in higher eukaryotes, our understanding on evolutionary conservation and biological relevance of the fundamental cellular process in a wide range of eukaryotes including protists are limited. Lipids function as the key sensor and recruiter of secondary effectors, regulate basic cellular processes such as cell proliferation, differentiation, cell adhesion, migration, endocytosis, and exocytosis [4]. Conventionally, endo- and exocytosis have been considered as two evolutionary conserved and distinct cellular processes, and employ specific and distinct molecular effectors [5,6]. Nevertheless, the cell can maintain a constant size by keeping a temporal, spatial coordination between endo- and exocytic pathways through some key coordinating molecules such as synuclein, intersectin, and sec4p, which were recently identified in higher eukaryotes [7–9]. Nonetheless, we still don't know whether coupling of endo and exocytosis is also evolutionary conserved in unicellular eukaryotes such as parasitic protozoa.

The unicellular parasitic protist *Entamoeba histolytica* is the etiological agent of human amoebiasis, which is endemic mostly in developing countries with considerable morbidity and mortality (infecting approximately 50 million people and resulting in 40 to 100 thousand deaths annually) [10–13]. In the course of evolution under ever-fluctuating host and natural environment, this protist has undergone remarkable alterations in the content and function of its sub-cellular compartments as well represented by its unique diversification of mitochondrion-related organelle, mitosome [14–16]. It also lacks well organized ER and Golgi apparatus [17]. Moreover, an active, motile trophozoite represents a dynamic synthesis and turnover of membranes and vesicles [18], in which the lipids have a fundamental role. Hence, we believe that *E. histolytica* is an appropriate system to study the evolutionary conservation and uniqueness of intra-cellular lipid trafficking in unicellular eukaryotes. Moreover, this enteric parasite vigorously exploits phagocytosis and trogocytosis (particularly trogocytosis, meaning internalization of a live cell by nibbling or chewing) and exocytosis as indispensable cellular processes for host tissue invasion and pathogenesis [19–22]. Thus, this organism is a suitable model system to study molecular machinery of these two distinct intracellular processes [23,24] thanks to the clearly visible nature of vesicular/vacuolar compartments and the availability of genetic and cell biological approaches.

Given the importance of lipids as common regulators of both endo- and exocytosis, we initiated our investigation on intra-cellular non-vesicular lipid trafficking mediated by lipid transfer proteins in *E. histolytica* with an aim to understand its molecular mechanisms and to identify the key coordinating molecules of these two fundamental cellular processes. The *E. histolytica* genome encodes a diverse repertoire of twenty two LTP homologs, with its large proportion (fifteen) belongs to steroidogenic acute regulatory protein (StAR)-related lipid transfer (START) proteins [25]. In this study, among fifteen START proteins, we chose two LTPs, EhLTP1 and EhLTP3 based on their relatively high mRNA expression in trophozoites of *E. histolytica* HM-1:IMSS cl6 strain [25,26]. We justified the selection with the assumption that the LTPs with relatively high mRNA expression are more likely essential for in vitro growth and survival. We found EhLTP3 is exclusively involved in trogocytosis of live host cells, phagocytosis of dead host cells, and endocytosis of the fluid phase marker, but not in exocytosis (herein, "exocytosis" includes recycling of endocytosed materials and biosynthetic secretion).

In contrast, EhLTP1 is involved in trogocytosis and phagocytosis, endocytosis of the fluid phase marker, and two modes of exocytosis. Importantly, EhLTP1 is also involved in pseudopod formation and migration. Furthermore, both EhLTPs are involved in cell proliferation and tissue destruction. Thus, these two LTPs play indispensable roles in two inter-connected and essential processes in this unicellular parasitic protist.

## Results

### Identification and domain organization of *E. histolytica* STARTs

We conducted an InterPro domain search (PFAM) analysis on AmoebaDB version 38 in order to identify the potential LTP candidates in *E. histolytica* HM-1:IMSS, as described in our previous review article [25]. The *E. histolytica* genome encodes a diverse repertoire of fifteen potential START homologs. Such a large number of START homologs suggest that *E. histolytica* heavily relies on START-mediated intracellular lipid trafficking. However, their relative mRNA expression levels significantly vary in *E. histolytica* as per AmoebaDB [26]. Three members of *E. histolytica* START protein homologs (EHI_080260, EHI_161070, and EHI_173480) showed relatively higher mRNA expression in HM-1:IMSS trophozoites compared to other START proteins [25]. We designates them as *E. histolytica* LTP (EhLTP) 1 (EHI_080260), EhLTP2 (EHI_161070), and EhLTP3 (EHI_173480), respectively. They possess only START domain (STARD) and lack any other associated domains or motifs [25]. In this study, we describe the biochemical and functional characterization of EhLTP1 and EhLTP3, while the investigation of EhLTP2 will be described elsewhere. Three other START protein homologs (EHI_182510, EHI_155260 and EHI_130730) showed lower mRNA expression in HM-1: IMSS. Such a spectrum of mRNA expression among START homologs possibly suggests that they are involved in various biological functions, depending on cellular and environmental conditions. Amino acid alignment of EhLTP1 and EhLTP3 by ClustalW (S1 Fig) indicates very low positional identity (12%). This indicates that their binding and transport specificities towards lipid ligands probably vary significantly. These two homologs (EhLTP1 and EhLTP3) also show very low (<20%) amino acid identities to their counterparts in human and yeast (S2 Fig), suggesting their significant divergence from those of human and other eukaryotes.

### EhLTP1 and EhLTP3 bind to phosphoinositides (PIs)

Recombinant EhLTP1 (rEhLTP1) and EhLTP3 (rEhLTP3) were expressed and the expression were verified by immunoblot analysis with anti-His antibody (S3D and S4D Figs). Lipid binding specificity of EhLTP1 and EhLTP3 was examined in a protein-lipid overlay assay. rEhLTP1 bound specifically with PtdIns, PI(4)P, PtdIns(4,5)P$_2$, and PtdIns(3,4,5)P$_3$, with the preference toward PI(4)P, followed by PtdIns(4,5)P$_2$ and PtdIns(3,4,5)P$_3$ (Fig 1A). An interaction of rEhLTP1 with cardiolipin and sulphatides was also observed (Fig 1A), which may be potentially due to their negative charge, but its relevance in vivo remains unknown. rEhLTP3 showed specific binding with PI(4)P, PtdIns(4,5)P$_2$, PtdIns(3,4,5)P$_3$, with the preference toward PtdIns(4,5)P$_2$ (Fig 1B). Weak binding to PA and sulphatides was also observed (Fig 1B). Neither rEhLTP1 nor rEhLTP3 bound to cholesterol, which is consistent with the premise that oxysterol-binding protein-related proteins [22] are likely involved in cholesterol transport. As a control, an irrelevant protein with the His tag (NAD kinase, rNADK) was used (Fig 1C).

### EhLTP1 and EhLTP3 have distinct PIs transfer activities

Since EhLTP1 and EhLTP3 bound mostly to PI(4)P, PtdIns(4,5)P$_2$, and PtdIns(3,4,5)P$_3$ in lipid binding assay as above (Fig 1), we further investigated their PI transport activities by in

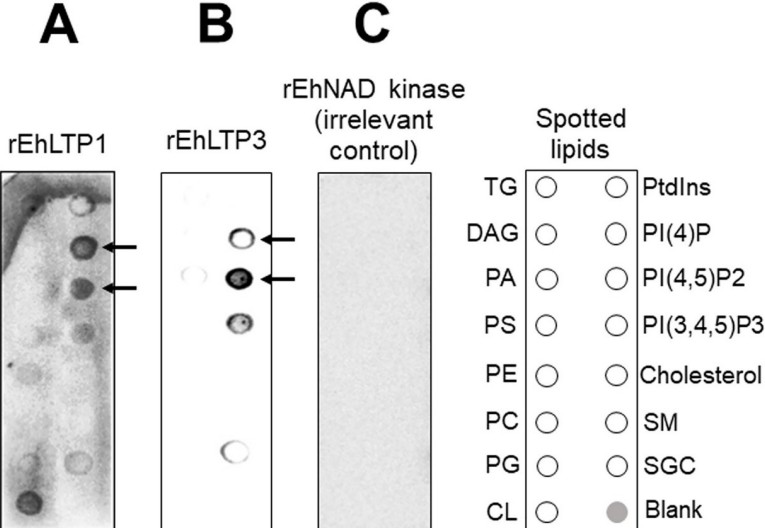

**Fig 1. Binding of recombinant EhLTP (rEhLTP) to lipids by protein lipid overlay assay.** (A) Membrane Lipid Strips™ were incubated overnight at 4˚C with 1 mg/ml of rEhLTP1. (B) Membrane Lipid Strips™ were incubated overnight at 4˚C with 1 mg/ml of rEhLTP3. Binding of protein to the lipid spots was detected by subsequent incubation with anti-His mouse monoclonal antibody and secondary anti-mouse peroxidase antibody as described in Methods. The lipids spotted on the strips are: Glyceryl tripalmitate or Triglyceride (GT), Diacylglycerol (DAG), Phosphatidic Acid (PA), Phosphatidylserine (PS), Phosphatidylethanolamine (PE), Phosphatidylcholine (PC), Phosphatidylglycerol (PG), Cardiolipin (CL), Phosphatidylinositol (PI), phosphatidylinositol 4-phosphate (PtdIns4P), phosphatidylinositol 4,5-bisphosphate [PtdIns(4,5)$P_2$], phosphatidylinositol 3,4,5-trisphosphate [PtdIns(3,4,5)$P_3$)], Cholesterol, Sphingomyelin (SM), 3-sulfogalactosylceramide or Sulfatide (SGC), and Blank. **C** Negative control, the same treatment as "**A**" or "**B**", the only exception was in place of study proteins (rEhLTP3 or rEhLTP1), we used rEhNAD kinase, which has no reported lipid binding ability.

vitro PI transport assay. We used PI(4)P and PtdIns(4,5)$P_2$ as representative PIs (Figs 2 and S5). In both assays using either PI(4)P or PtdIns(4,5)$P_2$, we observed a sharp increase in fluorescence after addition of rEhLTPs to DV (DV+LTP). This is consistent with the premise that Bodipy-PIs were extracted from DV and liberated in the assay solution by rEhLTP. Furthermore, additional approximately 2–3 fold increase in fluorescence was observed when AV was given to the assay mixture (DV+AV+LTP). However, no PIs transport was observed when rEhLTPs were not used (DV ONLY, DV+AV, and negative control). Lipid transfer activity was calculated using the formula described in Methods. These data provide direct biochemical evidence to support the notion that EhLTP1 and 3 have activity to extract PI(4)P and PtdIns(4,5)$P_2$ from donor liposomes and transfer to acceptor liposomes. rEhLTP1 mediated transport of PI(4)P slightly more efficiently (0.034 ± 0.0001% of total PtdIns4P / μg protein / hr) (Fig 2B) compared with PtdIns(4,5)$P_2$ (0.026 ± 0.001% of total PtdIns(4,5)$P_2$ / μg protein / hr) (S5B Fig). In contrast, rEhLTP3 mediates transport of PtdIns(4,5)$P_2$ (0.045 ± 0.001% of total PtdIns(4,5)$P_2$ / μg protein / hr) (S5D Fig) about 3 fold more efficiently than PI(4)P (0.014 ± 0.001% of total PI4,5)$P_2$ / μg protein / hr) (Fig 2D). These data provide direct biochemical evidence that EhLTP1 and 3 have activity to extract PI(4)P and PtdIns(4,5)$P_2$ from donor liposomes and transfer to acceptor liposomes; moreover, EhLTP3 has clear preference to PtdIns(4,5)$P_2$ over PI(4)P, while EhLTP1 has almost comparable activity towards PI(4)P and PtdIns(4,5)$P_2$.

## EhLTP1 and EhLTP3 are indispensable for *E. histolytica* growth

In order to investigate involvement and role of EhLTPs in various cellular processes in *E. histolytica*, we attempted to create cell lines in which the expression of *EhLTP1* or *EhLTP3* genes

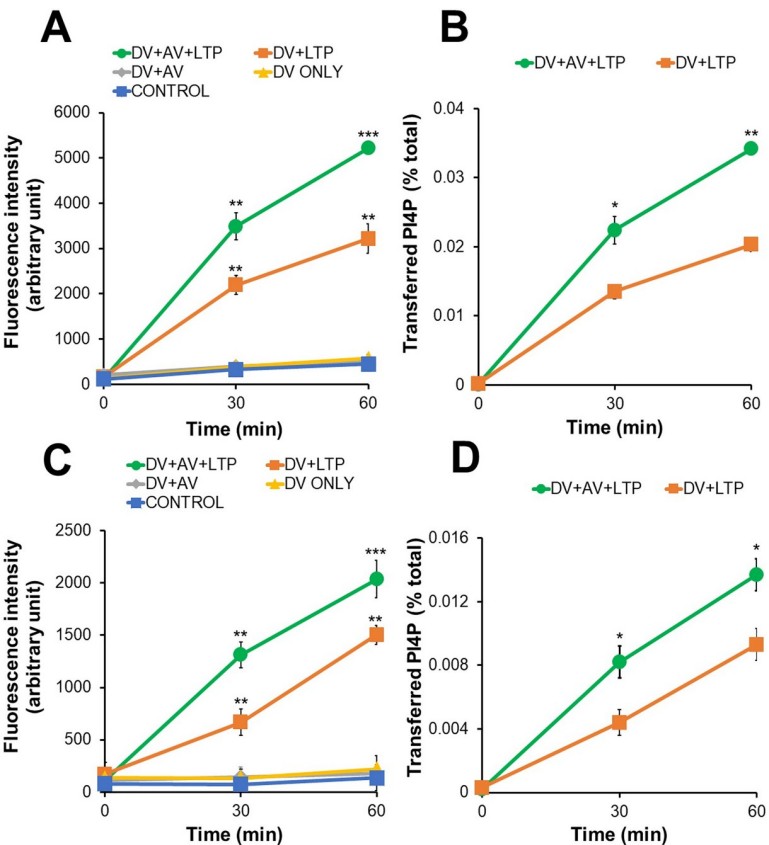

**Fig 2. In vitro transfer of PtdIns4P from donor to acceptor vesicles in the presence of rEhLTP1 (A,B) and rEhLTP3 (C,D).** Donor vesicles (DV) (1.5 μl) containing PtdIns4P were incubated in separate reaction without acceptor vesicles (AV) and recombinant proteins ("DV ONLY"), with 0.5 μg of either rEhLTP1 or rEhLTP3 ("DV +LTP"), with 3 μl of AV ("DV+AV"), or with 3 μl of AV and 0.5 μg of either rEhLTP1 or rEhLTP3 ("DV+AV+LTP") at room temperature. Fluorescence units were measured after 30 and 60 min of incubation. The percentage of transfer of PtdIns4P (B, D) were calculated from (A) and (C), respectively as described in Methods section. The experiments were conducted in duplicates three times (n = 3 with error bars indicating standard deviations). Statistical comparisons were made by Student's t-test (*P ≤ 0.05, **P ≤ 0.005, ***P ≤ 0.0005). The y axis represents arbitrary fluorescence units.

was repressed by antisense small RNA-mediated transcriptional gene silencing in G3 strain [27,28]. The specific and complete repression of the genes was confirmed in the gene silenced trophozoites in comparison to the transformant transfected with vector control (S3C and S4C Figs) by reverse transcriptase PCR of corresponding cDNA. Growth kinetics of the transformant lines where EhLTP1 (EhLTP1gs) or EhLTP3 (EhLTP3gs) was silenced was monitored. Both EhLTP1gs and EhLTP3gs showed remarkable growth retardation compared to vector control cell line (psAP2 vector) at 24–72 h post initiation of culture (S6A and S6B Fig). This strongly suggests the pivotal role of EhLTPs (EhLTP3 and EhLTP1) in *E. histolytica* trophozoite growth.

## EhLTP3, but not EhLTP1, is involved in endocytosis of the fluid-phase marker

Substantial growth retardation by *EhLTP1* and *EhLTP3* gene silencing indicates that these two LTPs are possibly involved in some fundamental nutritional acquisition processes. Since PIs are important signaling molecules involved in all stages of endocytosis and exocytosis

processes, particularly in vesicular trafficking, we first investigated the effect of gene silencing on fluid-phase endocytosis. Pinocytosis (endocytosis as measured by incorporation of fluid-phase markers) is a process used for the uptake of nutrients from the extracellular environment [21]. *E. histolytica* trophozoites in culture and in the host large intestine derive most of the nutrients through pinocytosis.

EhLTP3gs trophozoites showed substantial decrease (52.1% at 120 minutes) in endocytosis of the fluid-phase marker compared to psAP2 vector control (Fig 3B). On the contrary, the transformant line in which EhLTP3 was over-expressed as a GFP fusion protein (GFP-EhLTP3) showed 31.3% (approximately 1.32 fold) higher endocytic ability as compared to the vector control cell line (GFP control) at 60 min (Fig 3D). These results indicate the involvement and essentiality of EhLTP3 in fluid-phase endocytosis (Fig 3B and 3D). In contrast, gene silencing of *EhLTP1* led to no defect in endocytosis (Fig 3A). Nevertheless, EhLTP1 over-expressing line (GFP-EhLTP1) retained a slightly less amount of intra-cellular RITC-dextran as compared to vector control line at 60 and 120 min (8.5% and 14.1%, respectively) (Fig 3C). This may be attributable to a possibility that some portion of incorporated RITC-dextran was exocytosed from recycling endosomes (or "rapidly exchanging compartment") [22], indicating a possible role of EhLTP1 in exocytosis which was further verified in fluid phase exocytosis assay (see below).

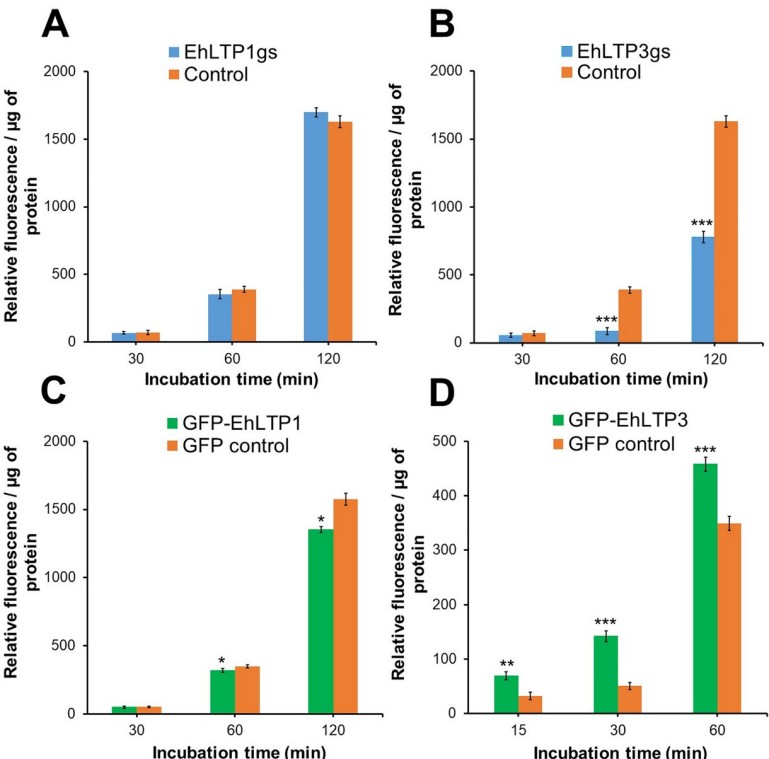

**Fig 3. Effects of gene silencing of *EhLTP1* (A) and *EhLTP3* (B) or overexpression of *EhLTP1* (C) and *EhLTP3* (D) on endocytosis of the fluid-phase marker RITC dextran.** *E. histolytica* gene silenced transformants [EhLTP1gs or EhLTP3gs, (A) or (B), respectively], overexpressing transformants [GFP-EhLTP1 or GFP-EhLTP3 in (C) or (D), respectively) and their corresponding vector control [psAP2 control (A, B) or GFP control (C, D)] cell lines were incubated with 2 mg ml$^{-1}$ of the fluid-phase marker RITC dextran in BI-S-33 medium for indicated time. Cells were collected, washed, and the fluorescence of incorporated RITC-dextran was measured using a fluorometer as per the method described in Methods. The experiment was repeated three times independently in duplicates (N = 3 with error bars indicating standard deviation). Statistical comparisons were made by Student's t-test ($^*P \leq 0.05$, $^{**}P \leq 0.005$, $^{***}P \leq 0.0005$).

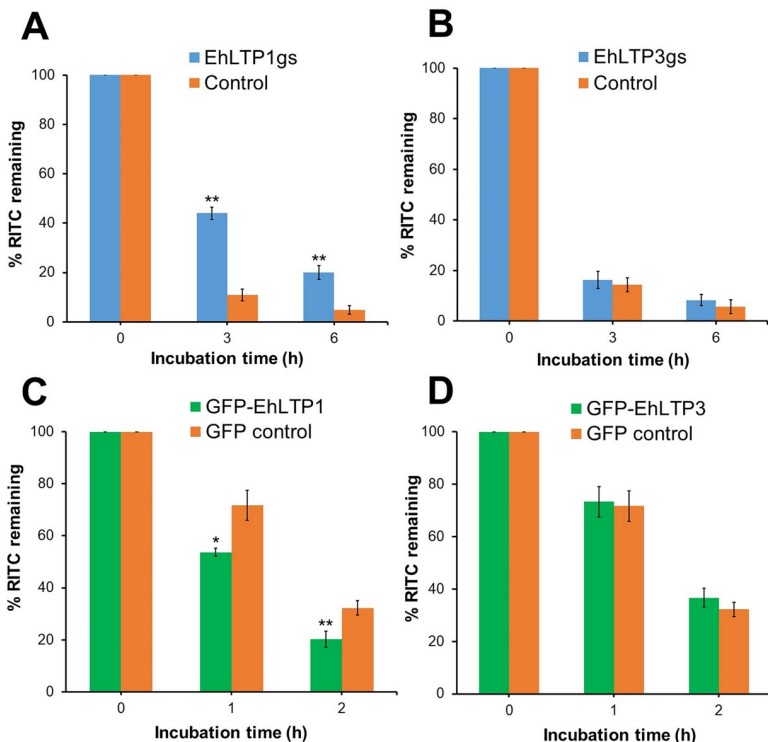

**Fig 4. Effects of gene silencing of *EhLTP1* (A) and *EhLTP3* (B) or overexpression of *EhLTP1* (C) and *EhLTP3* (D) on exocytosis of the fluid-phase marker RITC dextran.** *E. histolytica* gene silenced transformants [EhLTP1gs or EhLTP3gs, (A) or (B), respectively], overexpressing transformants [GFP-EhLTP1 or GFP-EhLTP3 (C) or (D), respectively] and their corresponding vector control [psAP2 control (A, B) or GFP control (C, D)] cell lines were incubated with 2 mg ml$^{-1}$ of the fluid-phase marker RITC dextran for 3 h, and then chased for indicated time. Cells were collected, washed, and the fluorescence of incorporated RITC-dextran was measured using a fluorometer as per the method described in Methods. The experiment was repeated three times independently in duplicates (N = 3 with error bars indicating standard deviation). Statistical comparisons were made by Student's t-test (*P $\leq$ 0.05, **P $\leq$ 0.005, ***P $\leq$ 0.0005).

## EhLTP1, but not EhLTP3, is involved in exocytosis of a fluid-phase marker and secretion of cysteine proteases

We investigated the effect of gene silencing on exocytosis of the fluid phase marker. The transformant cells were labeled with RITC-dextran for 3 h, washed with fresh BI-S-33 medium, and cultured in a RITC-dextran-free BI-S-33 medium. We measured the release of RITC-dextran by monitoring the decrease in the remaining intracellular RITC fluorescence. Trophozoites of two control amoebic lines (G3-derived strain transfected with psAP2 and HM-1-derived strain with pEhExGFP) exocytosed approximately 89 ± 2.39% or 68 ± 2.81% of incorporated RITC dextran in 3 or 2 h, respectively (Fig 4A and 4C). EhLTP1gs line was partially defective in exocytosis of RITC-dextran; it retained 33.2 ± 2.5% (4.07 fold more) RITC-dextran as compared to vector control line at 3 h (Fig 4A). Furthermore, EhLTP1 over-expressing line (GFP-EhLTP1) exocytosed 18.0 ± 1.6% (1.34 fold more) RITC-dextran into extracellular milieu compared to vector control (GFP control) line at 1 h (Fig 4C). In contrast, no change in fluid phase exocytosis was observed neither in EhLTP3 gene silenced (EhLTP3gs) or overexpressing (GFP-EhLTP3) lines compared to vector controls (Fig 4B and 4D). These observations indicate that EhLTP1, but not EhLTP3, plays an active role in exocytosis of the fluid phase marker.

We further investigated role of EhLTP1 on biosynthesis and secretion of cysteine proteases (CPs). It has been well established that *E. histolytica* trophozoites secrete CPs via biosynthesis and receptor (CPBF1) [29]-mediated secretion via the default secretory pathway. CPs and their secretory process play an indispensable role in amoebic invasion and tissue destruction due to their hydrolytic and degradative activities towards host cells and extracellular matrix proteins [30]. EhLTP1gs line had 127 ± 41.8% (2.27 fold) higher intra-cellular CP activity in cell lysates as compared to vector control (Fig 5A). In good agreement, the activity of extra-cellular CP secreted by EhLTP1gs line into the culture medium was decreased by 55.5 ± 9.79% (2.25 fold) as compared to vector control (Fig 5A). In contrast, EhLTP1 over-expressing line (GFP-EhLTP1) had 42.6 ± 3.59% (1.75 fold) less intra-cellular CP activity in cell lysates (Fig 5C), while it had 51.1 ± 6.70% (1.51 fold) higher extra-cellular CP activity in the culture supernatant as compared to vector control cell line (Fig 5C). These data clearly indicate that EhLTP1 is involved in the secretion of CP into the extra-cellular milieu through vesicular trafficking and secretion. Neither gene silencing (EhLTP3gs) nor overexpression (GFP-EhLTP3) of EhLTP3 caused any such alterations in both intra-cellular and extra-cellular CP activities compared to vector control (Fig 5B and 5D). These observations clearly suggest an isotype specific role of EhLTP1, but not EhLTP3, in both exocytosis of the fluid-phase marker and biosynthetic secretion of the lysosomal enzyme.

## EhLTP1, but not EhLTP3, is necessary for migration of *E. histolytica* trophozoites

We further investigated role of EhLTP1 and EhLTP3 in cell migration of *E. histolytica* trophozoites. Gene silencing of *EhLTP1* (EhLTP1gs) caused approximately 52.0 ± 7.22% (2.07 fold)

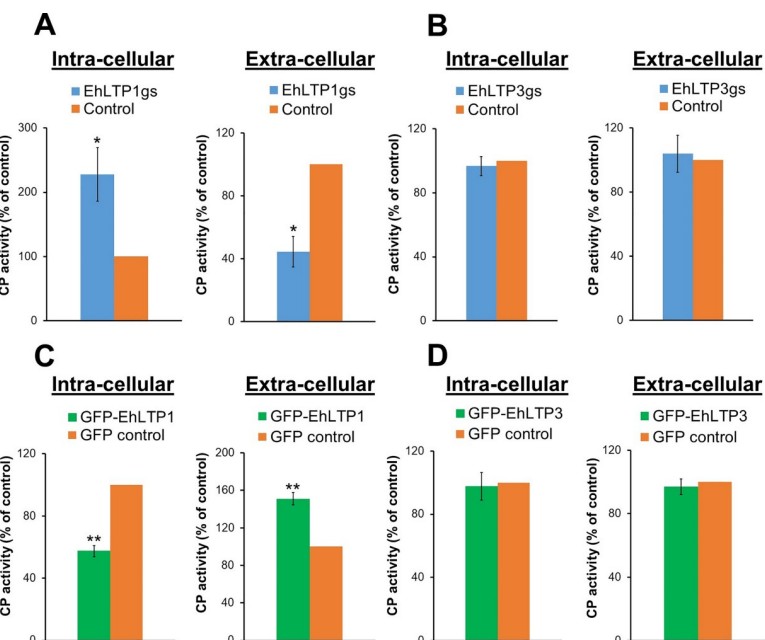

**Fig 5. Effects of gene silencing of *EhLTP1* (A) and *EhLTP3* (B) or overexpression of *EhLTP1* (C) and *EhLTP3* (D) on cysteine protease (CP) secretion by *E. histolytica* trophozoites.** The CP activity in the total lysate (intra-cellular CP activity) and culture supernatant (extra-cellular CP activity) of the transformants was measured as described in Methods section. The activities are shown as the percentage relative to the vector control transformant [psAP2 control (A, B) or GFP control (C, D)]. The experiment was repeated three times independently in duplicates (N = 3 with error bars indicating standard deviation). Statistical comparisons were made by Student's t-test (*P ≤ 0.05, **P ≤ 0.005, ***P ≤ 0.0005).

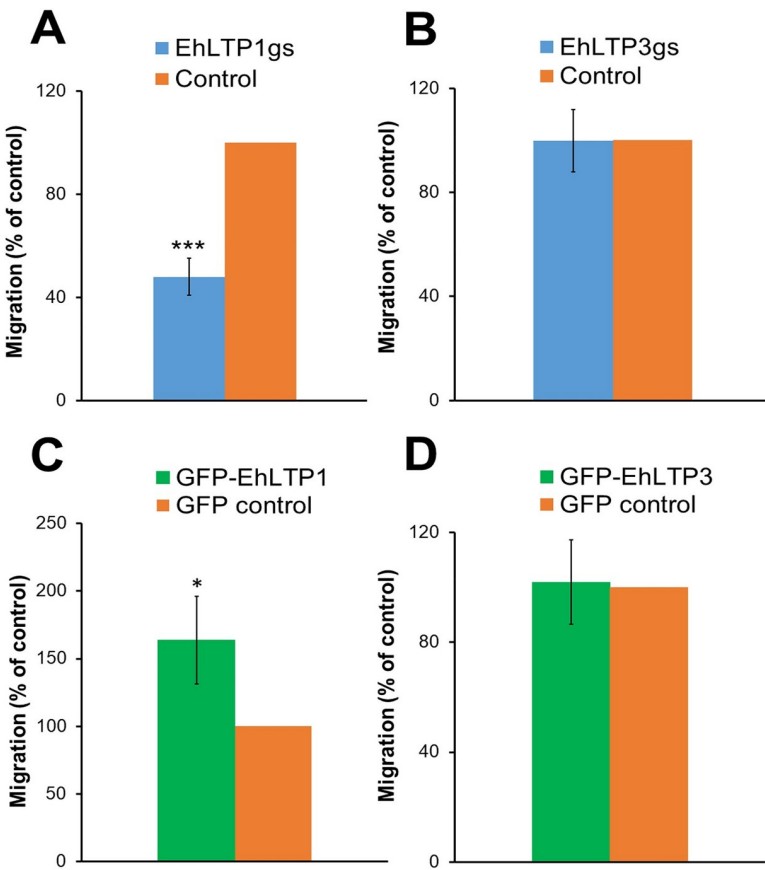

**Fig 6. Effects of gene silencing of *EhLTP1* (A) and *EhLTP3* (B) or overexpression of *EhLTP1* (C) and *EhLTP3* (D) on cell motility in the transwell migration assay of *E. histolytica* trophozoites.** Motility of *E. histolytica* transformants and vector control cell lines through 8 μm pores towards either BI-S-33 medium containing bovine serum or BI medium lacking serum (as Blank or gravity control) at 37˚C was assayed as described in Methods. The percentages of the cells that migrated from upper BI medium to lower BI-S-33 medium in 12h are shown. The experiment was repeated three times independently in duplicates (N = 3 with error bars indicating standard deviation). Statistical comparisons were made by Student's t-test (*P ≤ 0.05, **P ≤ 0.005, ***P ≤ 0.0005).

reduction in the motility of the cells when they were stimulated with bovine serum for 12 h at 37˚C compared to vector control line (psAP2) in the transwell migration assay (Fig 6A). In contrast, overexpression of EhLTP1 (GFP-EhLTP1) caused 63.7 ± 32.5% (1.63 fold) increase in motility towards stimuli compared to vector control (GFP control) at 6 hrs of incubation (Fig 6C). Gene silencing or overexpression of EhLTP3 exhibited no effect on the motility of *E. histolytica* trophozoites towards stimuli (Fig 6B and 6D). This clearly suggests the implication of EhLTP1 but not EhLTP3 in parasite's motility.

## Both EhLTP1 and EhLTP3 are essential for cytopathic activity of *E. histolytica* trophozoites

We further investigated whether the specific roles of EhLTP1 and EhLTP3 in endocytosis, exocytosis, CP secretion, and/or cell migration also contribute to the parasite's invasive and cytopathic properties. Destruction of monolayer of live CHO cells by *E. histolytica* trophozoites mimics the destruction of host cells and tissues by the parasite during infection. In order to reveal the role of EhLTP1 and EhLTP3 in this process, we labeled the live CHO cells by Cell-Tracker Blue and then incubated them with the *E. histolytica* lines where EhLTPs are

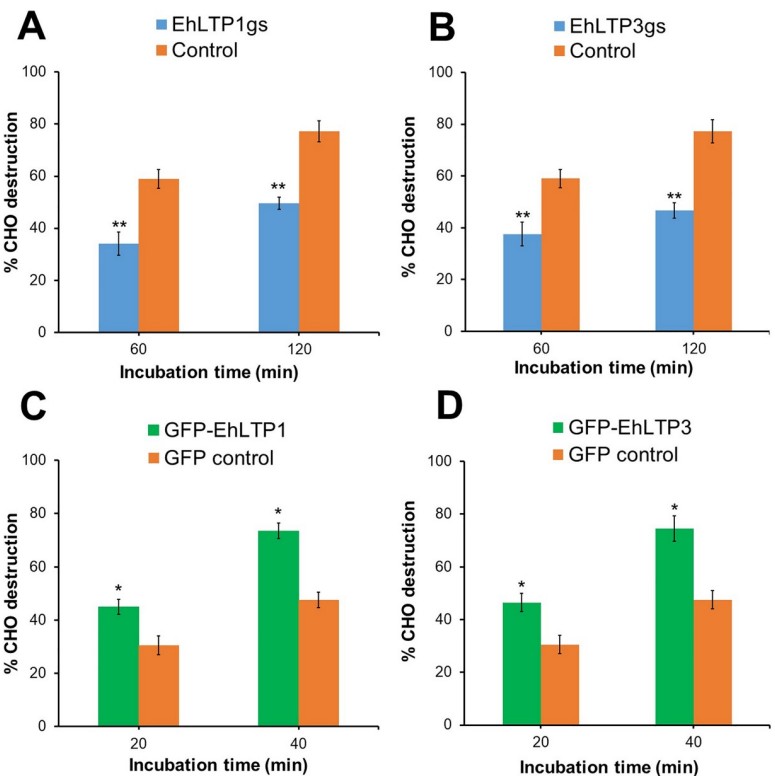

**Fig 7. Effects of gene silencing of *EhLTP1* (A) and *EhLTP3* (B) or overexpression of *EhLTP1* (C) and *EhLTP3* (D) on the cytopathy of *E. histolytica* trophozoites against mammalian cells.** Destruction of Cell Tracker Blue labelled CHO cells by *E. histolytica* transformants was assayed as described in Methods. The number of remaining adherent CHO cells was proportional to the intensity of Cell Tracker Blue signal and expressed as the percentage of the fluorescence of remaining untreated CHO cells. The experiment was repeated three times independently in duplicates (N = 3 with error bars indicating standard deviation). Statistical comparisons were made by Student's t-test (*P $\leq$ 0.05, **P $\leq$ 0.005, ***P $\leq$ 0.0005).

overexpressed or gene silenced (Fig 7). The destruction was inversely correlated with the remaining florescence of intact cells. Gene silencing of *EhLTP1* or *EhLTP3* caused reduction of cytolysis against CHO cells. In case of psAP2 vector control line, 77.3 ± 4.0% of CHO cells were destroyed by the trophozoites in 120 minutes (Fig 7A and 7B), while EhLTP1gs trophozoites destroyed only 49.6 ± 2.2% of CHO cells (28% reduction compared to control) (Fig 7A) and EhLTP3gs trophozoites destroyed only 46.6±3.0% of CHO cells (31% reduction compared to control) in the same duration (Fig 7B). Conversely, overexpression of EhLTP1 or EhLTP3 caused augmentation of cytolysis against CHO cells. In case of GFP control, about 47.5 ± 3.5% cells were destroyed in 40 minutes (Fig 7C and 7D), while EhLTP1- (GFP-EhLTP1) and EhLTP3- overexpressing cells (GFP-EhLTP3) destroyed 73.5 ± 2.9% (26% increase compared to control) and 74.5±4.9% (27% increase compared to control) of CHO cells, respectively in the same duration (Fig 7C and 7D). These results suggest that both of EhLTP1 and EhLTP3 influence the cytopathic activity of *E. histolytica* trophozoites.

## Cellular localization of EhLTP1 and EhLTP3 and their involvement in trogo- and phagocytosis in *E. histolytica*

It was well established that *E. histolytica* trophozoites employ two distinct endocytic modes to engulf mammalian cells. They initially kill the host cell in a contact dependent manner and then engulf the whole dead cell by phagocytosis [31], while they bite or chew fragments of the

live host cell by trogocytosis [19,23]. Although endocytosis and phagocytosis are well defined at the molecular level in model organisms [32,33], the regulation of trogocytosis is poorly understood [24,34]. As differentiation of these two processes is important to understand pathophysiology of amoebiasis, we next investigated by live imaging where EhLTP1 and EhLTP3 are located during phago- and trogocytosis, and how and when they participate in these endocytic mechanisms.

*E. histolytica* transformants expressing EhLTP1 or EhLTP3 with either GFP tag (GFP-EhLTP1 and GFP-EhLTP3, respectively) (S3B and S4B Figs) or HA tag (HA-EhLTP1 and HA-EhLTP3, respectively) at the amino terminus were produced (S3A and S4A Figs). The expression of proteins in *E. histolytica* trophozoites was verified by immunoblots with either anti-GFP or anti-HA antibody (S3A, S3B, S4A, and S4B Figs). Cellular localization of EhLTP1 and EhLTP3 under resting conditions and during trogocytosis and phagocytosis was examined in transformants expressing GFP tagged proteins.

## EhLTP1 and EhLTP3 are localized to the cytosol, and concentrated in newly forming pseudopods, and overexpression of EhLTP1 enhances pseudopod formation

Live imaging revealed that both of the proteins appear to be uniformly distributed in the cytosol (S1 and S4 Movies). No obvious difference in the distribution was observed between EhLTP1 and EhLTP3 (S1 and S4 Movies). However, during locomotive movement, these proteins seem to be enriched at newly forming pseudopods (S1 and S4 Movies). We also quantify the number of newly forming pseudopods in each transformant and vector control cell lines (S7 Fig), which indicates that GFP-EhLTP1 expressing trophozoites produced two fold more number of pseudopods (23.9± 4.5 μm per minute) compared to GFP-EhLTP3 (10.8± 2.5 μm per minute) and vector control lines (11.1± 2.8 μm per minute) (S7 Fig). This also supports our observation above that EhLTP1, but not EhLTP3, is involved in migration (Fig 6).

## Overexpression of EhLTP3 increases in the volume of endocytosis of the fluid-phase maker, but represses the number of invagination events

*E. histolytica* trophozoites overexpressing GFP-EhLTP3 showed notable membrane invagination at the plasma membrane ("the pinocytic cup" like structure) (S4 Movie). Such phenotype was less evident in *E. histolytica* trophozoites overexpressing GFP-EhLTP1 (S1 Movie) and in vector control cell line (S9 Movie). To see if increased endocytosis of RITC-dextran (see above) and augmented membrane invagination caused by EhLTP3 overexpression is attributable to increase in the number of endocytic events, we measured the invagination events per cell in 6.5 min (S8 Fig). Expression of GFP-EhLTP3 caused 46.4±0.9% reduction (mean±standard deviation, n = 3) in the number of invagination events compared to that in GFP control. In contrast, expression of GFP-EhLTP1 caused 30.2±9.1% increase in the number of invagination events compared to that in GFP control. These data, together with the quantitative data of endocytosed RITC-dextran, are consistent with the premise that GFP-EhLTP3 overexpression causes bigger and less frequent bites, whereas GFP-EhLTP1 overexpression causes smaller and slightly more frequent bites. Therefore, both EhLTP1 and EhLTP3 are involved in endocytosis of the fluid-phase marker, but in reciprocal manners. Overall, EhLTP3, and to a lesser extent, EhLTP1, are involved in fluid-phase endocytosis (Fig 3).

*Localization of EhLTP1 and EhLTP3 during trogocytosis.* Involvement of EhLTPs during trogocytosis of live mammalian cells by the *E. histolytica* trophozoites was also examined using live imaging. *E. histolytica* trophozoites nibble a small part of the live host cell during trogocytosis. In this process, trophozoites produce "pincers-like structure" followed by a narrow

tubular structure called "trogocytic tunnel" [19,20,23,35]. GFP-EhLTP1 appeared to be localized on the tip of the leading edge of newly forming "pincers-like structure" at relatively early time point (S2 Movie) and dissociated from the structure soon after (approximately 15 seconds after attachment), while GFP-EhLTP3 was localized to the inner lining of "the trogocytic tunnel", which connected the trogocytic mouth (that initiated at the attachment site) and the trogosome to be formed during trogocytosis (S5 Movie). The association of GFP-LTP3 and the trogocytic tunnel occurred relatively stably from the beginning of the formation of the tunnel until the full formation of the trogosomes. The localization of GFP-EhLTP1 and GFP-EhLTP3 during trogocytosis appeared to be largely overlapping in localization or temporarily comes one after another.

### EhLTP1 and EhLTP3 are recruited to the "pincers-like" structures (the apical portion) of the phagocytic cup during phagocytosis

*E. histolytica* trophozoites ingest dead host cells as a whole by phagocytosis [20]. Binding of the ligands on the dead cell surface to the potential receptor on the ameba cell surface, likely triggers reorganization of cytoskeleton underneath the binding site, resulting in the phagocytic cup formation [36]. A pair of tips of the phagocytic cup extend around the dead cell to engulf it, and finally fuse to each other, which leads to closure of the phagocytic cup and then formation of the phagosome. We examined how EhLTP1 and EhLTP3 are involved in phagocytosis of labeled pre-killed CHO cells by time lapse imaging. Both GFP-EhLTP1 and GFP-EhLTP3 were found to be recruited at the apical portion of the phagocytic cup (the "pincers-like" structures) and stay associated with the phagocytic cup untill closure of the phagocytic cup and scission of the phagosome (S3 and S6 Movies).

### EhLTP3 is recruited to the pinocytic site on the plasma membrane and the pinocytic cup during pinocytosis

Since our endocytosis assay (see above) indicated that only EhLTP3, but not EhLTP1, is involved in pinocytosis (endocytosis of fluid-phase markers) (Fig 3), we further investigated on the cellular localization of EhLTP3 during pinocytosis of RITC dextran in live imaging. GFP-EhLTP3 was recruited to the pinocytic site of a plasma membrane invagination during pinocytosis (S7 Movie), and associates with the pinocytic cup till the closure of the pinocytic cup (S7 Movie). In contrast, this recruitment to the pinocytic site was not observed for GFP-EhLTP1 (S8 Movie). These data further supported our premise that EhLTP3 is primarily involved in endocytosis of the fluid phase marker.

### Subcellular distribution of EhLTP1 and EhLTP3 by physical separation

Subcellular distribution was examined by density-dependent cellular fractionation of the transformants expressing HA-tagged EhLTPs. Cellular fractionation followed by immunoblot analysis revealed that HA-EhLTP1 and HA-EhLTP3 are present both in 100,000×g supernatant (s100) fraction corresponding to the cytosol and 100,000×g pellet (p100) fraction corresponding to the organelles (S3E and S4E Figs). The observed fractionation of EhLTP1 and EhLTP3 to the cytosolic fractions is consistent with our live imaging data (S1 and S4 Movies). However, a considerable amount of proteins are also associated with organelles.

### Reconfirmation of the involvement of EhLTP1 and EhLTP3 in trogocytosis and phagocytosis by gene silencing and overexpression

We further investigated the effects of gene silencing and overexpression of *EhLTP1* and *EhLTP3* on trogo- and phagocytosis. Gene silencing of *EhLTP1* (EhLTP1gs) or *EhLTP3*

(EhLTP3gs) caused reduction in the trogosome formation, i.e, the number of trogosomes formed per cell, compared to psAP2 control at all time points when amebae were fed with live CHO cells (S9A Fig). Similarly, EhLTP1gs and EhLTP3gs strains also showed a reduction in the number of phagosomes per cell when fed with prekilled CHO cells (S9B Fig). On the contrary, GFP-EhLTP1 or GFP-EhLTP3 expressing strain showed a higher number of trogosomes and phagosomes per cell, compared to mock GFP control at all time points when incubated with live or prekilled CHO cells, respectively (S9C and S9D Fig).

## Discussion

In our present study, we have demonstrated that LTPs are involved in trogo-, phago-, endo- (pino-), and exocytosis in *E. histolytica*. All of these processes require dynamic synthesis and turnover of lipid messengers and employ several lipid binding effectors such as PH (pleckstrin homology), PX (Phox homology), FYVE (Fab1, YOTB, Vac1, EEA1), PHD (plant homeodomain), C2, and ENTH (epsin N-terminal homology) domain containing proteins [20,23,33,35,37]. However, it remains largely unknown how intra-cellular lipid transport is regulated during these fundamental processes.

We have demonstrated that two key LTPs (EhLTP1 and EhLTP3) are coordinately involved in phago- and trogocytosis in *E. histolytica*. Importantly, the patterns of recruitment and distribution of EhLTP1 and EhLTP3 are markedly different depending upon the prey. Both proteins showed similar localization on the "apical portion" of the phagocytic cup during internalization of dead host cells. In contrast, in trogocytosis of a live mammalian cell, EhLTP1 was enriched on the tip of the leading edge of the newly forming "pincers-like" structure of the "trogocytic cup" at the site of attachment, on the initial stage of trogocytosis. Subsequently, EhLTP3 was recruited to the inner lining of "the trogocytic tunnel", which is formed after the initial attachment and formation of the trogocytic cup. Such isotype-specific and tempo-spatial recruitment of EhLTP1 and EhLTP3 was similar to that of AGC kinases 2 and 1, respectively [19], which may infer potential cross talk between the proteins via PIPs. The localization of EhLTP1 and EhLTP3, shown in this study, is similar with that of PtdIns(4,5)$P_2$ and their synthetic enzyme, EhPIPKI [20,38]. This is consistent with the fact that both EhLTP1 and 3 have PtdIns4P and PtdIns(4,5)$P_2$ transport activities. Hence, it is highly conceivable that EhLTP1 and 3 can facilitate the intracellular transport and consecutive recruitment of PtdIns4P and PtdIns(4,5)$P_2$ at those target sites during trogo- and phagocytic processes. They may also facilitate the transient in situ synthesis of PtdIns(4,5)$P_2$ followed by PtdIns(3,4,5)$P_3$)] via providing of the substrates [i.e. PtdIns4P and PtdIns(4,5)$P_2$)] to EhPIPKI and PtdIns 3-kinase, respectively (S9 Fig), as suggested as a role of LTPs in lipid metabolism [2]. The localization and dynamics of PIs including PtdIns4P [20,39], PtdIns3P [35], PtdIns (4,5)$P_2$ [38], PtdIns(3,4,5)$P_3$ [19,37] in *E. histolytica* during trogo- and phagocytosis were well studied through their binding effectors. It has been shown that EhFP4 interacts with PtdIns4P, EhSNX1 and 2 interact with PtdIns3P, and EhAGCK1 and 2 interact with PtdIns(3,4,5)$P_3$.

Gene silencing and overexpression of EhLTP1 affected phagocytosis more severely than trogocytosis, while manipulation of EhLTP3 influenced trogocytosis more than phagocytosis. Thus, EhLTP1 and EhLTP3 are engaged with trogo- and phagocytosis processes in different manners (S9 Fig). We have shown that EhLTP1 and EhLTP3 have clear preference towards PIs: EhLTP1 transports PI(4)P more efficiently than PtdIns(4,5)$P_2$, while EhLTP3 transports PtdIns(4,5)$P_2$ more efficiently than PI(4)P (Fig 1). EhLTP1, which has efficient PI(4)P transport activity, can trigger in situ synthesis of PtdIns(4,5)$P_2$ by EhPIPKI on the plasma membrane as discussed above. EhPIPKI and PtdIns(4,5)$P_2$ have been implicated in erythrophagocytosis [24], supporting the suggested preferred involvement of EhLTP1 in phagocytosis. On the other hand, EhLTP3 with efficient PtdIns(4,5)$P_2$ transport activity may

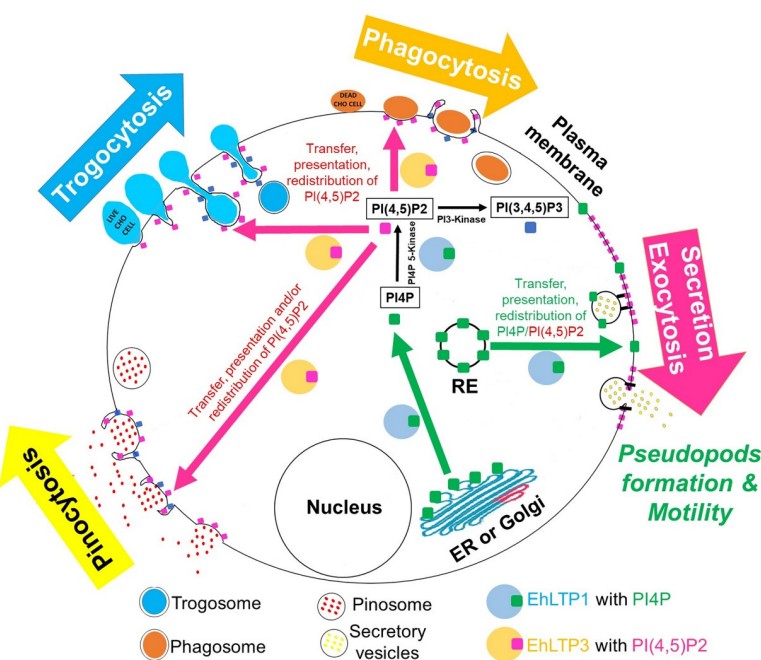

**Fig 8. Schematic diagram represents potential function of EhLTP1 and EhLTP3 in *E. histolytica* during endo-(trogocytosis, phagocytosis and pinocytosis) and exocytosis.** Both of EhLTP1 and EhLTP3 can transport PI(4)P to plasma membrane (PM) from endoplasmic reticulum (ER) or other cell organelles (for instances, RE, recycling endosome) and modulate the local synthesis of PIPs. However, EhLTP3 regulates exclusively the receptor-mediated endocytosis (trogocytosis, phagocytosis and pinocytosis) process via various mode of action of PITPs as proposed in the discussion. In contrast, EhLTP1 can regulates both receptor-mediated endocytosis (trogo- and phagocytosis) and exocytosis processes. It can facilitates the reorganization of membrane PtdIns(4,5)$P_2$ followed by priming and docking of the exocytic vesicle complex, hence the exocytosis.

promote in situ synthesis of PtdIns(3,4,5)$P_3$ by PtdIns 3-kinase, which may facilitate EhAGCK1 recruitment and trogocytosis. It is worth noting that EhLTP1 was previously identified from phagosomes isolated with beads [35], suggesting that EhLTP1 is involved in phagosome biogenesis from the initiation of phagocytosis to the maturation of phagosomes. In contrast, EhLTP3 may be involved only in early steps, relatively upstream of the cascade. This premise was further reinforced by other observations described below.

Further analysis based on phenotypic assays of genetically modified cell lines revealed that EhLTP1 but not EhLTP3 was involved in exocytic pathways, namely, fluid-phase exocytosis (Fig 4) and secretion of its importance virulence factor, CPs (Fig 5). EhLTP1 is the first example of LTP which functions in an intersection of endocytosis and exocytosis, although PtdIns(4,5)$P_2$ and PtdIns 3-kinase are implicated for coupling of endocytosis and exocytosis at the synapse of mammalian nervous system [40,41]. The priming and docking of the exocytic vesicle complex with the plasma membrane requires transient changes in the distribution of lipids [e.g., PtdIns(4,5)$P_2$] at the site of exocytosis [42]. EhLTP1 possibly facilitates such transient alteration in lipid distribution on the plasma membrane by extracting a lipid from or delivering it to a particular region of the membrane (Fig 8). In contrast, EhLTP3, not to EhLTP1, is involved in pinocytosis of the fluid-phase marker (Fig 3). EhLTP3 overexpression caused extensive membrane invagination ("the pinocytic cup" like structure) on the plasma membrane, suggestive of potential upregulation of the signal for membrane invagination by EhLTP3. The functional implication of PtdIns(4,5)$P_2$, PtdIns(3,4,5)$P_3$, and their biosynthetic enzymes such as PtdIns 3-kinase, PTEN in macropinocytosis was reported in both *Dictyostelium* and mammals [43]. The involvement of PIs in fluid-phase pinocytosis of *E. histolytica*

was previously shown [19], where we showed that EhAGCK2, one of the PtdIns(3,4,5)P$_3$ effectors, is involved in pinocytosis. However, the membrane invagination to form the "pinocytic cup" during macropinocytosis requires local enrichment and transient synthesis of PIs [specifically, PtdIns(4,5)P$_2$ and PtdIns(3,4,5)P$_3$] on the plasma membrane followed by actomyosin reorganization. LTP has been reported as an essential components of lipid biosynthetic machinery, can facilitates local lipid metabolism of a targeted organelle by providing the lipid substrate to biosynthetic enzymes and also by removal of its product from the site of synthesis [2]. It is conceivable that overexpression of EhLTP3 accelerates transient synthesis of PtdIns (4,5)P$_2$, followed by PtdIns(3,4,5)P$_3$ synthesis by PtdIns 3-kinase on the plasma membrane (Fig 8). This in turn facilitates the actomyosin reorganization, membrane invagination and 'pinocytic cup' formation, thus regulates the pinocytic process.

EhLTP1 overexpression induced pseudopods formation and migration, indicating that EhLTP1, but not EhLTP3, is involved in the parasite's motility. Cell motility involves local production and degradation of PtdIns(3,4,5)P$_3$ at the plasma membrane, resulting in a net accumulation of PtdIns(3,4,5)P$_3$ at the leading edge [44]. This ultimately leads to actin polymerization and formation of pseudopods. The interplay between PtdIns 3-kinase and PTEN regulates the levels of PtdIns(4,5)P$_2$ and PtdIns(3,4,5)P$_3$, which act on downstream effectors, including the Rac-GTPase, Arf-GTPases, Akt kinase (PKB) and pleckstrin homology (PH) domain proteins (PhdB, PhdG) have been shown to regulate the motility by controlling the actin cytoskeleton in *Dictyostelium* and mammals [44]. Similarly, EhPIPKI, PtdIns 3-kinase, and their corresponding products, PtdIns(4,5)P$_2$ and PtdIns(3,4,5)P$_3$, respectively, regulate the actin dynamics, and control motility of *Entamoeba* trophozoites [37,38]. The intracellular levels of PtdIns(4,5)P$_2$ inversely correlates with motility [45], while PtdIns(3,4,5)P$_3$, localized on extending pseudopods positively regulates the migrating ability of *Entamoeba* trophozoites [37]. The effectors of PtdIns 3-kinase pathway such as, EhRho1 [45], Ehcoactosin [46], and PH domain containing proteins [37] also regulate cell motility. EhLTP1 with PtdIns4P and PtdIns(4,5)P$_2$ transfer activities can potentially facilitates the transient synthesis of PtdIns(3,4,5)P$_3$ by PtdIns 3-kinase as mentioned above (Fig 8), which in turn regulates the actin dynamics, pseudopods formation and motility. LTP have been implicated in cytoskeleton organization and migration of cancer cell line through MAPK and PtdIns 3-kinase/AKT signaling pathways [47,48].

PtdIns4P and PtdIns(4,5)P$_2$ transport activity possess specific roles in pinocytosis, trogocytosis, phagocytosis, exocytosis (biosynthetic and recycling), and motility in *E. histolytica*. EhLTP1 is exclusively involved in pseudopod formation and motility while EhLTP3 is in the fluid-phase endocytosis. Both EhLTP1 and EhLTP3 play non-overlapping roles in trogocytosis and phagocytosis of mammalian cells. Taken together, the present study also demonstrates novel biological implications of LTPs in endocytic pathways such as pinocytosis, trogocytosis, and phagocytosis, for which there was no precedent in unicellular eukaryotes with respect to their previously suggested roles in multicellular eukaryotes [49,50]. Moreover, we have shown a single LTP for the first time that regulates endocytosis, phago-/trogocytosis, and biosynthetic secretion/exocytosis. Our data should help our understanding of unique evolution of lipid transfer mechanisms in eukaryotes and may also provide a potential target for the development of new therapeutics against eukaryotic pathogens.

## Methods

### Organisms and culture

Trophozoites of *E. histolytica* strain HM-1:IMSS cl-6 were cultured axenically at 35°C in 6mL screw-capped Pyrex glass tubes or plastic culture flasks in BI-S-33 medium as previously described [51]. For the detection of proteins in the lysates and endocytic assays, approximately

$1X10^6$ trophozoites of the late-logarithmic growth phase were cultivated in 15 ml of BI-S-33 medium under anaerobic conditions using Anaerocult A (Merck, Darmstadt, Germany) on a 90-mm culture plate at 35°C for 2 h. CHO cells were grown in F-12 medium (Invitrogen-Gibco) supplemented with 10% fetal bovine serum on a 10-cm-diameter tissue culture dish (IWAKI, Tokyo, Japan) under 5% CO2 at 37°C. *Escherichia coli* strain DH5α was purchased from Life Technologies (Tokyo, Japan). All chemicals of analytical grade were purchased from Sigma-Aldrich (Tokyo, Japan) unless otherwise stated.

## Plasmid construction

Standard techniques were used for routine DNA manipulation, subcloning and plasmid construction as previously described [52]. The protein coding regions of EhLTP3 and EhLTP1 were amplified by PCR from cDNA using specific oligonucleotides containing appropriate restriction sites (S1 Table). The PCR-amplified DNA fragments were digested with SmaI and XhoI, and ligated into SmaI and XhoI restriction sites of either pEhExHA expression vector (protein tagged with three tandem copies of the influenza virus haemagglutinin /HA peptide) or pEhExGFP expression vector (protein tagged with Green Florescent Protein/GFP) [20]. To produce full length histidine (His) tagged recombinant protein, the PCR-amplified DNA fragments were digested with BamHI and SalI, and ligated into BamHI and SalI restriction sites of pCOLD I expression vector (Takara Bio Inc, Japan) according to previous report [53]. For gene silencing of EhLTP3 and EhLTP1, the 420bp long 5'-end of the protein coding region of corresponding genes were amplified by PCR from cDNA using sense and antisense oligonucleotides. The PCR amplified DNA fragment was digested with StuI and SacI, and ligated into StuI- and SacI-digested psAP2-Gunma [54]. The gene-silenced strains were established by the transfection of G3 strain [55] with the corresponding plasmids as described below.

## Bacterial expression and purification of recombinant EhLTP (rEhLTP)

The pCOLD1-EhLTP3 and pCOLD1-EhLTP1 expression constructs were introduced into competent *E. coli* BL21 (DE3) cells by heat shock at 42°C for 30 s, and the resulting transformants were grown at 37°C in 100 ml of Luria Bertani medium in the presence of 50 μg/ml ampicillin. The overnight culture was then used to inoculate 500 ml of fresh medium, which was further cultured at 37°C with shaking at 180 rpm. When the $A_{600}$ reached 0.6, 1 mM isopropyl β-D-thiogalactopyranoside was added to induce protein expression, and cultivation was continued for 24 h at 15°C. The *E. coli* cells were then harvested by centrifugation at 4000 × *g* for 20 min at 4°C, and the resulting cell pellet was washed with PBS (pH 7.4) and re-suspended in 20 ml of lysis buffer (50 mM Tris-HCl (pH 8.0), 300 mM NaCl, and 10 mM imidazole) containing 0.1% Triton X-100 (v/v), 100 μg/ml lysozyme, and 1 mM phenylmethylsulfonyl fluoride. After a 30-min incubation at room temperature, the cells were sonicated on ice and centrifuged at 25,000 × *g* for 15 min at 4°C. The supernatant was mixed with 1.2 ml of a 50% nickel-nitrilotriacetic acid His-bind slurry (Qiagen, Tokyo, Japan) and incubated for 1 h at 4°C with gentle shaking. The rEhLTP-bound resin was washed three times with buffer A (50 mM Tris-HCl (pH 8.0), 300 mM NaCl, and 0.1% Triton X-100, v/v) containing 10–50 mM imidazole, and bound proteins were then eluted with buffer A containing 100–300 mM imidazole. The integrity and purity of the rEhLTP proteins were confirmed by 12% SDS-PAGE analysis and Coomassie Brilliant Blue staining and also by immunoblot analysis with anti-His antibody (Cell signaling technology, US) The concentrations of the purified proteins were spectrophotometrically determined by the Bradford method using bovine serum albumin as a standard as previously described [56]. The rEhLTP proteins were stored at −80°C in 20% glycerol in small aliquots until needed.

## Protein-lipid overlay assay

Previously expressed and purified His tag recombinant EhLTP3 (rEhLTP3) and EhLTP1 (rEhLTP1) were used to study their abilities to bind with lipid molecules as per previously described protocol with slight modifications [57]. Briefly, each Membrane Lipid Strip (P-6002, Echelon Bioscience Inc.) where the following lipids were spotted: triglyceride (GT), diacylglycerol (DAG), phosphatidic Acid (PA), phosphatidylserine (PS), phosphatidylethanolamine (PE), phosphatidylcholine (PC), phosphatidylglycerol (PG), cardiolipin (CL), phosphatidylinositol (PtdIns), PtdIns 4- phosphate [PI(4)P] (PtdIns phosphate called phosphoinositide, designated as PI), PtdIns 4,5-bisphosphate [PtdIns(4,5)$P_2$], PtdIns 3,4,5-trisphosphate [PtdIns (3,4,5)$P_3$], cholesterol, sphingomyelin (SM), 3-sulfogalactosylceramide or sulfatide (SGC), and Blue Blank; was blocked with 3% BSA in PBS for 2h at RT prior to incubation with either rEhLTP3 or rEhLTP1 (1mg/mL) in PBS with 3% BSA for 16 h at 4˚C. rEhNADH kinase [58] was used as negative control (since, no report was available regarding its lipid binding ability). After the strip had been washed three times with PBS containing 0.1% Tween 20 (PBS-T) buffer, it was incubated at RT with 1:1000 anti-His mouse monoclonal antibody (Thermo-Fisher Scientific, USA) in PBS with 3% BSA for 2 h, then washed and incubated at RT with 1:6000 HRP-conjugated goat anti-mouse IgG (Thermo-Fisher Scientific, USA) in PBS with 3% BSA for 1 h. Blots were washed and developed by chemiluminescence (PerkinElmer).

## In vitro lipid transport assay using rEhLTP

Phosphatidylcholine (PC) was purchased from Avanti Lipids (Alabaster, AL) and Bodipy-labelled PtdIns4P and PtdIns(4,5)$P_2$ were purchased from Echelon Biosciences Inc (Salt Lake City, US). The phospholipid transfer assay was performed as per the previously described protocol [59]. The only modification was replacement of NBD (nitrobenzoxadiazole)-labeled lipids with Bodipy-labeled lipids. The donor vesicle (DV) was composed of 450 nmol/ml of PC and >2 molar % of Bodipy-PtdIns4P or PtdIns(4,5)$P_2$ (note that the quenching concentration of Bodipy-labeled lipids was 2 molar % [60]). The acceptor vesicle (AV) was composed of 2,400 nmol/ml of PC. The vesicles were prepared as described previously [59]. The assay is based on the transfer of Bodipy-PIs from DV to AV in the presence of rEhLTP (either rEhLTP3 or rEhLTP1) (S10 Fig). DV prepared with Bodipy-PIs at self-quenching concentration has no or minimal fluorescence. AV contained only PC. Increase in fluorescence was detected only when Bodipy-PIs were extracted from DV by rEhLTP and present in assay solution, which was monitored as a function of time. The assay was done in black, clear bottom 96-well microtiter plates (Thermo Scientific, US). In the sample reaction ("DV+AV+LTP"), we added 1.5μl of the DV, 3μl of the AV, 10μl assay buffer (10mM Tris-HCl buffer, pH 7.5, 2mM EDTA, 150mM NaCl and 0.1% BSA) and purified rEhLTP (0.5–1.0 μg) in 100μl assay solution in triplicate. Along with each sample reaction we also ran four different types of control reactions. "DV ONLY": 1.5μl of the DV incubated in assay buffer without AV and rEhLTP; "DV+AV" (also represents as BLANK): 1.5μl of the DV and 3ul of the AV in assay buffer without rEhLTP; "DV+LTP": 1.5μl of the DV in assay buffer with rEhLTP; "Negative control" represent same as sample reaction (DV+AV+LTP) but in place of rEhLTP we used any irrelevant recombinant protein which do not have any known lipid transfer ability. In our assay we used one recombinant metabolic enzyme of *E. histolytica* (rEhNADH kinase [58]) as an irrelevant protein. Plates were incubated at room temperature for different time periods and read with a fluorescence plate reader (SpectraMax Paradigm Multi-Mode Detection Platform, Molecular Devices, US) using 485 nm excitation and 525 nm emission wavelengths. Total fluorescence in DV was determined by adding 98.5μl of isopropanol to 1.5μl of the DV. Lipid transfer activity (percentage transfer) was calculated by the following equation: percentage transfer = (arbitrary

fluorescence unit in assay well—blank value) / (total fluorescence unit—blank value) ×100. The specific activity is expressed as the percentage of transfer per micrograms of protein per hour.

## Production of *E. histolytica* transformants

Transfection of *E. histolytica* was conducted as described. Briefly, approximately $10^5$ trophozoites were seeded onto 35-mm diameter wells of a six-well culture plate and incubated at 35˚C for 1 h. The LipofectAMINE plasmid DNA complexes were prepared in OPTI-MEM I medium (Life Technologies) supplemented with 5 mg/ml L-cysteine and 1 mg/ml ascorbic acid (transfection medium) and pH 6.8. Transfection medium containing 5µg of the plasmid was mixed with 10 µl of LipofectAMINE PLUS (Life Technologies) and kept at room temperature for 15 min. This mixture was combined with 20µg (10µl) of LipofectAMINE, kept at room temperature for 15 min, diluted with 0.5 ml of transfection medium, and added to the seeded trophozoites after removing BI-S-33 medium. The plate was then incubated at 35˚C for 4 h. After incubation with the LipofectAMINE-DNA complex, 70–90% of the trophozoites were viable. The trophozoites were transferred to fresh BI-S-33 medium and further cultivated at 35˚C for 24 h. G418 was then added to the cultures at 1 µg/ml initially and increased to 10 µg/ml gradually by increasing at the rate of 1 µg/ml every 24h. Finally transformants were maintained at 10 µg/ml of G418 in the medium.

## Preparation of whole lysates from transformants and immunoblot analysis

Approximately $1 \times 10^6$ transformant trophozoites were harvested 48 h after initiation of culture and washed with 2% glucose in 1X phosphate buffer saline (PBS) three times. Cells were counted and resuspended in 500 µL homogenization buffer (50mM Tris-HCl, pH 7.5, 250mM sucrose, 50mM NaCl and 0.5 mg/ml E-64 (Peptide Institute, Osaka, Japan) protease inhibitor). Cells were disrupted mechanically by a Dounce homogenizer and kept on ice for 30 min with intermittent vortexing followed by centrifugation at 500×g for 30 min at 4˚C for removing the insoluble cellular debris. The supernatant, representing total cell lysate, was carefully collected. The expression of N-terminal GFP-tagged full length wild type proteins in *E. histolytica* trophozoites were verified by immunoblots with anti-GFP mouse monoclonal antibody (Abcam, USA), while expression of N-terminal HA-tagged full length wild type proteins were verified with anti-HA mouse monoclonal antibody (Thermo-Fisher Scientific, USA).

## Monitoring of growth kinetics

Trophozoite cultures were continuously maintained in mid-log phase and growth kinetics has been performed. Briefly, trophozoite cultures were placed on ice for 5 min to detach cells from the glass surface. Cells were collected by centrifugation at 500×g for 5 min at room temperature. After discarding the spent medium, the pellet was re-suspended in 1 mL of BI-S-33 medium. Cell densities were estimated on a haemocytometer. Approximately 10,000 trophozoites were inoculated in 6 mL fresh BI-S-33 medium. Cultures were examined every 24 h for 3 days.

## Measurement of fluid-phase endocytosis

The fluid-phase endocytosis assay of *E. histolytica* trophozoites were performed as described previously [19]. Briefly, $5 \times 10^5$ amoebic transformants were incubated in BI-S-33 medium containing the fluorescent fluid-phase marker RITC-dextran (2 mg ml$^{-1}$; MW = 70 000; Sigma-Aldrich, Japan) at 35˚C for indicated time points. The labeled cells were collected, washed

three times with ice-cold PBS. The cell pellets were then suspended in 300 μl of 50 mM Tris-HCl, pH 7.0 containing 1% Triton-X100 and vortexed for 15 s. Fluorescence intensity was measured using a fluorometer (F-2500, Hitachi, Japan) at excitation and emission wavelengths of 570 and 610 nm respectively.

### Measurement of fluid-phase exocytosis

The fluid-phase exocytosis assay of *E. histolytica* trophozoites were performed as described previously [22]. Briefly, $5X10^5$ amoebic transformants were incubated in BI-S-33 medium containing the fluorescent fluid-phase marker RITC-dextran (2 mg ml$^{-1}$; MW = 70 000; Sigma-Aldrich, Japan) at 35˚C for 3 h to saturate all endocytic compartments with the marker. The labelled cells were collected, washed and re-suspended in warm marker-free BI-S-33 medium, and incubated at 35˚C for 6 h (for gene silenced strain and psAP2 vector control cell line) or 2 h (for overexpressor and GFP vector control cell line). At specific time points (0 h, 3 h and 6 h), cells were collected and washed three times with ice-cold PBS. The cell pellets were then suspended in 300 μl of 50 mM Tris-HCl, pH 7.0 containing 1% Triton-X100 and vortexed for 15 s. Fluorescence intensity was measured using a fluorometer (F-2500, Hitachi, Japan) at excitation and emission wavelengths of 570 and 610 nm respectively.

### Assay for cysteine protease activity

Cysteine protease activity was measured with the cleavage of the synthetic peptide substrate z-Arg-Arg-7-amino-4-trifluoromethylcoumarin (ICN, Aurora, OH) monitored by a spectrophotometric method as described previously [61,62]. The activities were expressed in mmol of z-Arg-Arg-7-amino-4-trifluoromethylcoumarin produced per milligram of lysate protein.

### Transwell migration assay

Transwell migration assay of amoebic transformants was performed as previously described with slight modifications. Briefly, a 100 μl/well suspension of $3 \times 10^4$ amoebic transformants in BI medium (medium without serum) was distributed on top of 6.5-mm transwell inserts (upper chamber) with polycarbonate membranes containing 8 um pores (Thermo Fisher Scientific, USA). In the lower chamber, 600 μl of either BI-S-33 medium (medium with serum) (serum in medium acts as stimulus for trophozoite migration) or BI medium (as Blank or gravity control) was placed. The parasites were then incubated at 37˚C for 12h, after which the plate was placed on ice for 15 min, and cells that migrated to the bottom chamber were harvested and counted with a haemocytometer.

### Determination of cytopathic activity

The destruction of CHO monolayers was quantified as described previously [63] with slight modifications. Briefly, CHO cells were labelled with 40 μM of Cell tracker blue dye (Molecular probes, Eugene, OR) in F12 medium containing 10% FCS. After staining, labelled CHO cells were washed three times with fresh pre-warmed F12 medium. Approximately, $1.5 \times 10^5$ *E. histolytica* transformants were mixed with pre-warmed OPTI-MEM I medium (Life Technologies) supplemented with 5 mg/ml L-cysteine and 1 mg/ml ascorbic acid (transfection medium) pH 6.8, and added over labelled CHO cells and incubated at 35˚C for 0, 60 and 120 min. After incubation, the remaining CHO cells were collected using trypsin and the fluorescence of Cell tracker blue was measured using a fluorometer (F-2500, Hitachi, Japan) with excitation and emission at 353 and 465 nm respectively. The number of adherent CHO cells

was proportional to the intensity of Cell tracker blue staining and expressed as a percentage of the remaining fluorescence of untreated CHO cells.

### Live cell imaging

Approximately $5\times10^5$ transformants were cultured on a 35mm collagen-coated glass- bottom culture dish (MatTek Corporation, Ashland, MA) in 3 ml of BI-S-33 medium under anaerobic conditions. CHO cells were stained for 30 min with 40 μM Cell tracker blue dye (Molecular probes, Eugene, OR) in F12 medium containing 10% FCS. After staining, CHO cells were washed three times with fresh F12 medium, and approximately $2 \times 10^5$ CHO cells in 200 μl F12 medium were added to the GFP tagged protein expressing amoeba in a glass-bottom dish. Transfection medium (TM) [OPTI-MEM I medium (Life Technologies) supplemented with 5 mg/ml L-cysteine and 1 mg/ml ascorbic acid] containing the fluorescent fluid-phase marker RITC-dextran (2 mg ml$^{-1}$; MW = 70 000; Sigma-Aldrich, Japan) were added to the GFP tagged protein expressing amoeba to study the pinocytosis process in live *E. histolytica* trophozoites. The culture was carefully covered with a coverslip, and overloaded medium was removed. The junction of the coverslip and slide glass was sealed with nail polish, and the culture was incubated at 35˚C in a temperature control unit on Zeiss, LSM780 equipped with a 63x/1.4 oil immersion objective and CCD camera.

### Quantification of trogo- and phagocytosis of *E. histolytica* transformants

Approximately $1\times10^5$ transformants were cultured on a 35mm collagen-coated glass-bottom culture dish (MatTek Corporation, Ashland, MA) in 3 ml of BI-S-33 medium under anaerobic conditions. Live (for trogocytosis assay) or pre-killed (for phagocytosis assay) CHO cells were labelled with 40 μM Cell tracker orange dye (Molecular probes, Eugene, OR) as described above. CHO cells were then washed three times with fresh F12 medium, and $3X10^5$ CHO cells in 200 μl BIS-33 medium were added to the amoeba transformants and incubated for 0, 15, 30 or 60 min. After incubation, the CHO cells were removed and the amoeba were washed with warm PBS. The cells were then fixed with 3.7% paraformaldehyde (PFA) and observed in LSM780 equipped with a 63x/1.4 oil immersion objective. A total of thirty-five cells from each microscopic field (a total of seven microscopic field) of each transformants were randomly selected in each independent experiment and the total number of trogosomes or phagosomes in those cells were counted at indicated time points. The results were expressed as average number of trogosomes or phagosomes per cell.

### Cell fractionation and immunoblot analysis

Subcellular fractionation has been performed as described previously [64]. Briefly, trophozoites of the amoeba transformants expressing HA -EhLTP3 and HA-EhLTP1, and the mock transformant, transfected with empty HA vector, were washed three times with PBS containing 2% glucose. After resuspension in homogenization buffer (50mM Tris-HCl, pH 7.5, 250mM sucrose, 50mM NaCl and 0.5 mg/mL E-64 protease inhibitor), cells were disrupted mechanically by a Dounce homogenizer on the ice, centrifuged at 500×g for 5 min, and the supernatant was collected to remove unbroken cells. The supernatant fraction was centrifuged at 5000×g for 10 min to isolate pellet (p5) and supernatant fractions (s5). The 5000×g supernatant fraction (s5) was further centrifuged at 100,000×g for 60 min to produce a 100,000×g supernatant (s100) and pellet (p100) fractions. The pellets at each step were further washed twice with homogenization buffer and re-centrifuged at 100,000×g for 10 min to minimize carryover. Immunoblot analysis was performed using the fractions and anti-HA mouse monoclonal antibody. Anti-CPBF1 (cysteine protease binding family protein 1) [29] and anti-CS1

(cysteine synthase 1) [65] rabbit antisera were used as organelle membrane and cytosolic markers, respectively.

## Statistical analysis

The statistical analysis of the data was done by using Graph pad software. The statistical comparisons were performed by using unpaired *Student*'s t-test.

## Supporting information

**S1 Fig. Amino acid sequence alignment of EhLTP1 and EhLTP3 by ClustalW.** Red and Green boxes indicate the "STARD" domain of EhLTP1 and EhLTP3, respectively.
(TIFF)

**S2 Fig. Percentage identities of protein sequences of START homologs from human (STARD1-15 from humans), *E. histolytica* EhLTP1 (EHI_080260) and EhLTP3 (EHI_173480).** Identities below 20 percent are shown as '<20'. Note that both EhLTP1 and EhLTP3 are highly diverse from their counterparts from humans (identities were below 20 percent).
(TIFF)

**S3 Fig. Validation of expression of epitope-tagged EhLTP1.** (A) Expression of HA-tagged full length EhLTP1 (HA-EhLTP1) in *E. histolytica* trophozoites detected by anti-HA antibody. (B) Expression of GFP-tagged full length EhLTP1 (GFP-EhLTP1) in *E. histolytica* trophozoites detected by anti-GFP antibody. (C) The expression of EhLTP1 was silenced by antisense small RNA-mediated transcriptional gene silencing in G3 strain of *E. histolytica*. The expression was monitored by reverse transcriptase PCR. Lane 1, EhLTP1gs; Lane 2 vector control line (psAP2-Gunma). The RNA polymerase II gene was used as loading control. The silencing of amoebapore gene in transfected G3 strain was also checked. (D) His tag recombinant EhLTP1 (rEhLTP1) was expressed and purified. Expression of recombinant protein (rEhLTP1) was verified by CBB and also by immunoblot analysis with anti-His antibody. Lane 5, rEhLTP1 (approximately 27kD) indicated by an arrow. (E) Sub-cellular fractionation of transformant expressing HA-EhLTP1 followed by immunoblot analysis. Sub-cellular fractionation was performed as described in Methods section. Immunoblot analysis was performed using anti-HA mouse monoclonal antibody (1:1000). Anti-CPBF1 (cysteine protease binding family protein 1) (1:500) and anti-CS1 (cysteine synthase 1) (1:500) rabbit antisera were used as organelle membrane and cytosolic markers, respectively. Lane 1: marker, lane 2: 5000×g pellet fraction (p5), lane 3: 5000×g supernatant fraction (s5), lane 4: 100,000×g pellet fraction (p100), lane 5: 100,000×g supernatant fraction (s100) and lane 6: total lysate (TL).
(TIFF)

**S4 Fig. Validation of expression of epitope-tagged EhLTP3.** (A) Expression of HA-tagged full length EhLTP3 (HA-EhLTP3) in *E. histolytica* trophozoites detected by anti-HA antibody. (B) Expression of GFP-tagged full length EhLTP3 (GFP-EhLTP3) in *E. histolytica* trophozoites detected by anti-GFP antibody. (C) The expression of EhLTP3 was silenced by antisense small RNA-mediated transcriptional gene silencing in G3 strain of *E. histolytica*. The expression was monitored by reverse transcriptase PCR. Lane 1, EhLTP3gs; lane 2, vector control line (psAP2-Gunma). The RNA polymerase II gene was used as loading control. The silencing of amoebapore gene in transfected G3 strain was also verified. (D) His tag recombinant EhLTP3 (rEhLTP3) was expressed and purified. Expression of recombinant protein (rEhLTP3) was verified by CBB and also by immunoblot analysis with anti-His antibody. Lane 4, rEhLTP3 (approximately 29 kD), indicated by an arrow. (E) Sub-cellular fractionation of transformant

expressing HA-EhLTP3 followed by immunoblot analysis. Sub-cellular fractionation was performed as described in Methods section. Immunoblot analysis was performed using anti-HA mouse monoclonal antibody (1:1000). Anti-CPBF1 (cysteine protease binding family protein 1) (1:500) and anti-CS1 (cysteine synthase 1) (1:500) rabbit antisera were used as organelle membrane and cytosolic markers, respectively. Lane 1: marker, lane 2: 5000×g pellet fraction (p5), lane 3: 5000×g supernatant fraction (s5), lane 4: 100,000×g pellet fraction (p100), lane 5: 100,000×g supernatant fraction (s100) and lane 6: total lysate (TL).
(TIFF)

**S5 Fig. In vitro transfer of PtdIns(4,5)P$_2$ from donor to acceptor vesicles in the presence of rEhLTP1 (A, B) and rEhLTP3 (C, D).** Donor vesicles (DV) (1.5 μl) containing PtdIns(4,5)P$_2$ were incubated in separate reaction without acceptor vesicles (AV) and recombinant proteins ("DV ONLY"), with 0.5 μg of either rEhLTP1 or rEhLTP3 ("DV+LTP"), with 3 μl of AV ("DV+AV"), or with 3 μl of AV and 0.5 μg of either rEhLTP1 or rEhLTP3 ("DV+AV+LTP") at room temperature. Fluorescence units were measured after 30 and 60 min of incubation. The percentage of transfer of PtdIns(4,5)P$_2$ (B, D) were calculated from (A, C), respectively as described in Methods. The experiments were conducted in duplicates three times (n = 3 with error bars indicating standard deviations). Statistical comparisons were made by *Student*'s t-test (*P $\leq$ 0.05, **P $\leq$ 0.005, ***P $\leq$ 0.0005). The y axis represents arbitrary fluorescence units.
(TIFF)

**S6 Fig. Growth of the *E. histolytica* strains in which *EhLTP1* (A) or *EhLTP3* gene (B) was silenced.** Gene kinetics of EhLTP1gs (A), EhLTP3gs (B), and their mock transformant (with psAP2 control) during 72 h cultivation in BI-S-33 medium is shown. The experiment was repeated three times independently in duplicates (N = 3 with error bars indicating standard deviations). Statistical comparisons were made by Student's t-test (*P $\leq$ 0.05, **P $\leq$ 0.005, ***P $\leq$ 0.0005).
(TIFF)

**S7 Fig. Effects of overexpression of *EhLTP1* and *EhLTP3* on pseudopod formation by *E. histolytica* trophozoites.** The number of newly forming pseudopods were quantified in *E. histolytica* overexpressing transformants (GFP-EhLTP1 or GFP-EhLTP3) and their corresponding vector control (GFP control) cell lines in a specified time. The results were expressed as average number of pseudopods per cell per min. The experiment was repeated three times independently in duplicates (N = 3 with error bars indicating standard deviations). Statistical comparisons were made by Student's t-test (*P $\leq$ 0.05, **P $\leq$ 0.005, ***P $\leq$ 0.0005).
(TIFF)

**S8 Fig. Effects of overexpression of GFP-EhLTP1 and EhLTP3 on the number of invaginations during endocytosis of the fluid-phase maker.** A total of ten to fifteen GFP-positive cells of each transformants were randomly selected each in three independent experiments incubated with BI-S-33 medium containing 2 μg/ml RITC-dextran. The time-lapse images were captured at 2.0 s intervals for 6.5 mins. The number of invagination was counted and expressed as average number of invaginations per cell per minute in percentage relative to that in GFP control. The experiment was repeated three times independently (N = 3 with error bars indicating standard deviation). Statistical comparisons were made by *Tukey* test (*P $\leq$ 0.02).
(TIFF)

**S9 Fig. Effects of gene silencing (A, B) and overexpression (C, D) of *EhLTP1* and *EhLTP3* on trogocytosis (A, C) and phagocytosis (B, D) efficiency of *E. histolytica* trophozoites.** Cell

Tracker Orange labelled CHO cells were co-incubated with *E. histolytica* transformants for indicated time as described in Methods. A total of thirty-five cells from each microscopic field (a total of seven microscopic field) of each transformants were randomly selected in each independent experiment and the total number of trogosomes or phagosomes in those cells were counted at indicated time points. The results were expressed as average number of trogosomes or phagosomes per cell. The experiment was repeated three times independently in duplicates (N = 3 with error bars indicating standard deviation). Statistical comparisons were made by Student's t-test (*P ≤ 0.05, **P ≤ 0.005, ***P ≤ 0.0005).
(TIFF)

**S10 Fig. A schematic diagram for the principle of in vitro lipid transfer assay.**
(TIFF)

**S1 Table. A list of primers used in this study.**
(TIFF)

**S1 Movie. A movie showing the localization of GFP-EhLTP1 (green) in live *E. histolytica* trophozoites.** A number of pseudopods in different directions can be visualized. The enrichment of GFP-EhLTP1 in newly forming pseudopods can be observed.
(AVI)

**S2 Movie. A movie showing the localization of GFP-EhLTP1 (green) in live *E. histolytica* trophozoites during trogocytosis of live CHO cells (blue).** GFP-EhLTP1 was localized on the tip of the leading edge of newly forming "pincers-like structure" during initiation of trogocytosis.
(MOV)

**S3 Movie. A movie showing the localization of GFP-EhLTP1 (green) in live *E. histolytica* trophozoites during phagocytosis of dead CHO cells (blue).** GFP-EhLTP1 was found to be recruited at the "apical portion" of the phagocytic cup (or the "pincers-like structures") and stay associated with the phagocytic cup till closure of the phagocytic cup and scission of the phagosome.
(MOV)

**S4 Movie. A movie showing the localization of GFP-EhLTP3 (green) in live *E. histolytica* trophozoites.** The extensive membrane invagination in the cell periphery (formation of "pinocytic cup" like structures in cell periphery) in live *E. histolytica* trophozoites can also be observed.
(AVI)

**S5 Movie. A movie showing the localization of GFP-EhLTP3 (green) in live *E. histolytica* trophozoites during trogocytosis of live CHO cells (blue).** GFP-EhLTP3 was localized to the inner lining of "the trogocytic tunnel" during trogocytosis.
(MOV)

**S6 Movie. A movie showing the localization of GFP-EhLTP3 (green) in live *E. histolytica* trophozoites during phagocytosis of dead CHO cells (blue).** GFP-EhLTP3 was found to be recruited at the "apical portion" of the phagocytic cup (or the "pincers-like structures") and stay associated with the phagocytic cup till closure of the phagocytic cup and scission of the phagosome.
(MOV)

**S7 Movie. A movie showing the localization of GFP-EhLTP3 (green) in live *E. histolytica* trophozoites during pinocytosis of RITC dextran (red).** GFP-EhLTP3 was recruited to the pinocytic site of a plasma membrane invagination during pinocytosis, and associates with the pinocytic cup till the closure of the pinocytic cup.
(MOV)

**S8 Movie. A movie showing the localization of GFP-EhLTP1 (green) in live *E. histolytica* trophozoites during pinocytosis of RITC dextran (red).** GFP-EhLTP1 was not associated with the pinocytic cup and pinosomes.
(MOV)

**S9 Movie. A movie showing the localization of GFP protein in *E. histolytica* trophozoites expressing empty vector control (i.e. GFP control).**
(MOV)

## Acknowledgments

We thank Fuyuki Tokumasu, Nagasaki University for his help on lipid transfer assay. We also thank all members of Nozaki lab for technical assistance and discussions.

## Author Contributions

**Conceptualization:** Tomoyoshi Nozaki.

**Funding acquisition:** Tomoyoshi Nozaki.

**Investigation:** Koushik Das, Natsuki Watanabe, Tomoyoshi Nozaki.

**Methodology:** Koushik Das, Natsuki Watanabe, Tomoyoshi Nozaki.

**Project administration:** Tomoyoshi Nozaki.

**Resources:** Tomoyoshi Nozaki.

**Supervision:** Tomoyoshi Nozaki.

**Validation:** Koushik Das, Natsuki Watanabe, Tomoyoshi Nozaki.

**Writing – original draft:** Koushik Das, Tomoyoshi Nozaki.

**Writing – review & editing:** Natsuki Watanabe, Tomoyoshi Nozaki.

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
