## [Decision Letter · Decision Letter 0]

18 Jan 2021

Dear Prof. Nozaki,

Thank you very much for submitting your manuscript "Two StAR-related lipid transfer proteins play specific roles in endocytosis, exocytosis, and motility in the parasitic protist Entamoeba histolytica" for consideration at PLOS Pathogens. As with all papers reviewed by the journal, your manuscript was reviewed by members of the editorial board and by several independent reviewers. The reviewers appreciated the attention to an important topic. Based on the reviews, we are likely to accept this manuscript for publication, providing that you modify the manuscript according to the review recommendations.

Sincerely,

William A. Petri, Jr.

Associate Editor

PLOS Pathogens

Vern Carruthers

Section Editor

PLOS Pathogens

Kasturi Haldar

Editor-in-Chief

PLOS Pathogens

orcid.org/0000-0001-5065-158X

Michael Malim

Editor-in-Chief

PLOS Pathogens

orcid.org/0000-0002-7699-2064

Reviewer Comments (if any, and for reference):

Reviewer's Responses to Questions

**Part I - Summary**

Reviewer #1: In this study, the authors characterise the function of the lipid transfer proteins (LTPs) EhLPT1 and EhLPT2 in detail. E. histolytica possesses 15 StAR related lipid transfer (START) domain-containing proteins, with EhLTP1 and EhLTP3 being highly expressed in cell culture.

Both EhLTP1 and EhLTP3 are able to take up certain phosphatidylinositols from liposomes and release them to other liposomes.

For further characterisation, silencing and overexpressing transfectants were produced. Growth is inhibited by silencing the expression of both genes. Silencing of EhLTP3 expression leads to a reduction in endocytosis, while EhLTP1 has no effect on endocytosis but is involved in exocytosis. Furthermore, EhLTP1 is involved in the secretion of cysteine peptidases, probably playing a role in vesicle trafficking. It is also involved in cell migration. None of this is true for EhLPT3. Both molecules have an influence on cytopathic activity and are involved in phagocytosis and trogocytosis.

Reviewer #2: On the basis of their previous work which identified 22 potential homologs of lipid transfer proteins in Entamoeba histolytica, the authors were interested in two proteins having lipid binding characteristics because the genes encoding these proteins are well expressed under the culture conditions of the parasite.

E. histolytica is a pathogenic amoeba which has a very dynamic system of renewal of its internal and surface membranes. The two proteins of interest (LTP1 and LTP3) could be good markers of this membrane dynamics. In this objective the project is sound.

**Part II – Major Issues: Key Experiments Required for Acceptance**

Reviewer #1: This is a very cleanly conducted study that will be of interest to a wide readership. I do have one comment, however. There is, at least theoretically, the possibility that the production of transfectants leads to off target effects. Thus, there is a slight possibility that the determined partly different activities of both molecules are due to the corresponding off target effects. Therefore, it is important to determine the expression profile of the silencer and overexpression transfectants using next generation sequencing (RNAseq).

Minor point

Reviewer #2: Strengths of the work

Use of strains blocked in the synthesis of LTP1 and LTP3 by gene silencing or on the contrary) overexpressing the coding genes; the authors first demonstrate that LTP1 and LTP3 bind to lipids (mainly phosphoinositides) and furthermore, they conducted numerous tests and determine that LTP1 is involved in exocytosis and motility of the parasite while LTP3 is necessary for endocytosis. Both proteins are involved in amoebic cytotoxicity to mammalian cells, although cysteine proteinases (believed to be cytotoxicity factor) are mainly secreted under LTP1 activity. Finally, the two proteins (following distinct recruitment patterns) are also involved in the trogocytosis of living mammalian cells and the phagocytosis of dead cells.

All these numerous performed phenotypic tests were conducted in clean fashion, with the appropriate controls, and the data provide a solid background that can be used in mechanistic studies on the dynamics of PI trafficking and membrane dynamics, a question that should be addressed in further studies.

Principal weaknesses

1. The steroidogenic acute regulatory (STAR) protein is present in phospholipid transfer proteins and plays a major role in the supply of cholesterol to mitochondria. LTP1 and LTP3 in E. histolytica (an organism lacking mitochondria) contain the STAR domain and appear to be deleted from other lipid binding domains. Unfortunately, the presentation of LTP1 and LTP3 in this article is very poor. In the absence of amoeba STAR structural data, which exists from other organisms, there are many approaches to research the characteristics of STAR and other domains in these candidate proteins in E. histolytica. In addition, the authors do not discuss the absence of signal for cholesterol (or ceramide very abundant in the amoeba) following their lipid overlay test (figure 1) and also the fact that LTP1 binds to cardiolipin is not discussed.

2. In the liposome experiment (which was the only mechanistic investigation of LTPs in this work), LTP1 and LTP3 have activity to extract PI (4) P and PtdIns (4,5) P2 from donor liposomes and transfer them to acceptor liposomes. The authors stated that LTP3 prefers PtdIn (4,5) P2 over PI (4) P, while LTP3 has a weak but reversed preference for PI (lane 169). For this reader, the data in the figures 2 A and B are quite similar to C and D since PtdIns (4,5) P2 as a donor is not specified.

3. In line 313, the authors argue that the overexpression of LTP3 causes an increase in invagination of the plasma membrane. Looking carefully at the videos this phenotype is not obvious. In video 1 (LTP1), I notice invaginations in images 340, 435, 483. The number of invaginations per cell must be counted in an equivalent period of time.

4. The characteristics described in the text (line 352) are not observed in the video microscopy 7. These data need precision. Pynosomes are visible but not colocalization with RITC dextran particles. Control using either the LTP1 construct and / or WT cells may help to narrow the observation

5. The localization dynamics of PIs in E. histolytica during trogo- and phagocytosis has been extensively studied through their binding effectors, including several proteins studied by the author’s team. There is a long speculation in the discussion on this point; however, there is no information on the potential interaction of LTP with these effectors either by colocalization experiments (using their commercially available constructs and antibodies identifying PIs) and / or in their discussion.

**Part III – Minor Issues: Editorial and Data Presentation Modifications**

Reviewer #1: The number of E. histolytica infections and deaths has increased dramatically in recent years due to improved hygienic conditions. Unfortunately, there are only limited recent figures for this. (GBD 2013 Mortality and Causes of Death Collaborators, Global, regional, and national age-sex specific all-cause and cause-specific mortality for 240 causes of death, 1990-2013: a systematic analysis for the Global Burden of Disease Study 2013, Lancet 2015

Reviewer #2: Concerning the manuscript body

-The material and methods section must follow the same organization (order) as the results section

- From line 86. There is a confusing notion which, in my opinion, equates all phenomena based on ingestion of something with endocytosis; but endocytosis is a well-defined mechanistic cellular process. I think trogocytosis is a different process from endocytosis and that they cannot be included as an example of endocytosis, it's confusing. Even phagocytosis, which in ancient literature appears to be an endocytic process, is now well defined because mechanically speaking, it does not follow the endocytic pathways in the early stages. Moreover, no one knows if the pieces of human cells taken by the amoeba are ingested by endocytosis. I recommend to read these two recent articles PMID: 30504135 to PMID: 32366574 which clearly allows to conclude in distinction of these concepts. Even reference 43 in your text makes this difference. Please edit the text along the manuscript with this distinction in mind.

-Line 100. Strain c6 is not a reference strain. Please correct.

-The discussion is too long, dispersed and should gain in interest if a concordance between the literature and the results can be established (see comment 3). The diagram shown in Figure S10 might help the main text. You may want to consider placing it in the main article.

-The assessment of line 536 "may also provide a potential target for the development of new therapies against eukaryotic pathogens" is greatly overstated as it is not clear how new therapies against amoebiasis might include LTPs.

PLOS authors have the option to publish the peer review history of their article (what does this mean?). If published, this will include your full peer review and any attached files.

Reviewer #1: No

Reviewer #2: No
---

## [Editor Report · Decision Letter 1]

9 Apr 2021

Dear Prof. Nozaki,

We are pleased to inform you that your manuscript 'Two StAR-related lipid transfer proteins play specific roles in endocytosis, exocytosis, and motility in the parasitic protist Entamoeba histolytica' has been provisionally accepted for publication in PLOS Pathogens.

Best regards,

William A. Petri, Jr.

Associate Editor

PLOS Pathogens

Vern Carruthers

Section Editor

PLOS Pathogens

Kasturi Haldar

Editor-in-Chief

PLOS Pathogens

orcid.org/0000-0001-5065-158X

Michael Malim

Editor-in-Chief

PLOS Pathogens

orcid.org/0000-0002-7699-2064
---

## [Editor Report · Acceptance letter]

22 Apr 2021

Dear Prof. Nozaki,

We are delighted to inform you that your manuscript, "Two StAR-related lipid transfer proteins play specific roles in endocytosis, exocytosis, and motility in the parasitic protist Entamoeba histolytica," has been formally accepted for publication in PLOS Pathogens.

Best regards,

Kasturi Haldar

Editor-in-Chief

PLOS Pathogens

orcid.org/0000-0001-5065-158X

Michael Malim

Editor-in-Chief

PLOS Pathogens

orcid.org/0000-0002-7699-2064